# Proteomics Characterization of Food-Derived Bioactive Peptides with Anti-Allergic and Anti-Inflammatory Properties

**DOI:** 10.3390/nu14204400

**Published:** 2022-10-20

**Authors:** Ana G. Abril, Manuel Pazos, Tomás G. Villa, Pilar Calo-Mata, Jorge Barros-Velázquez, Mónica Carrera

**Affiliations:** 1Department of Microbiology and Parasitology, Faculty of Pharmacy, University of Santiago de Compostela, 15898 Santiago de Compostela, Spain; 2Department of Food Technology, Spanish National Research Council (CSIC), Marine Research Institute (IIM), 36208 Vigo, Spain; 3Department of Analytical Chemistry, Nutrition and Food Science, Food Technology Division, School of Veterinary Sciences, University of Santiago de Compostela, Campus Lugo, 27002 Lugo, Spain

**Keywords:** proteomics, anti-allergic peptides, anti-inflammatory peptides, bioactive peptides, food-derived bioactive peptides, mass spectrometry

## Abstract

Bioactive peptides are found in foods and dietary supplements and are responsible for health benefits with applications in human and animal medicine. The health benefits include antihypertensive, antimicrobial, antithrombotic, immunomodulatory, opioid, antioxidant, anti-allergic and anti-inflammatory functions. Bioactive peptides can be obtained by microbial action, mainly by the gastrointestinal microbiota from proteins present in food, originating from either vegetable or animal matter or by the action of different gastrointestinal proteases. Proteomics can play an important role in the identification of bioactive peptides. High-resolution mass spectrometry is the principal technique used to detect and identify different types of analytes present in complex mixtures, even when available at low concentrations. Moreover, proteomics may provide the characterization of epitopes to develop new food allergy vaccines and the use of immunomodulating peptides to induce oral tolerance toward offending food allergens or even to prevent allergic sensitization. In addition, food-derived bioactive peptides have been investigated for their anti-inflammatory properties to provide safer alternatives to nonsteroidal anti-inflammatory drugs (NSAIDs). All these bioactive peptides can be a potential source of novel drugs and ingredients in food and pharmaceuticals. The following review is focused on food-derived bioactive peptides with antiallergic and anti-inflammatory properties and summarizes the new insights into the use of proteomics for their identification and quantification.

## 1. Introduction

The National Institutes of Health (NIH), the main government agency responsible for biomedical and public health in the USA, defines the term bioactive peptides as “compounds that are constituents in foods and dietary supplements, other than those needed to meet basic human nutritional needs, which are responsible for changes in health status” [1,2]. These compounds can originate either from animals (i.e., casein protein) or plants (i.e., polyphenols). They can also be produced by microorganisms, both prokaryotes and lower eukaryotes, such as *Penicillium* spp. and *Talaromyces* spp. [3], and they can also include metabolic products generated by the action of microorganisms on either animal or plant proteins in environments such as the gastrointestinal tract of animals [4]. These bioactive peptides display health benefits and, as such, constitute an integral part of ‘nutraceuticals’, a term coined by DeFelice in 1989 that incorporates the words ‘nutrition’ and ‘pharmaceutical’ [5]. Nutraceuticals are described by the Merriam-Webster dictionary as “a foodstuff (such as a fortified food or dietary supplement) that provides health benefits in addition to its basic nutritional value”. The term nutraceutical may be considered as indicated in the book by Hasler in 2005 [6]. However, the regulation varies from country to country. At present, there is no global regulatory network to govern these compounds, with nutraceutics falling into different categories depending on the country. For example, while they are considered “foods in general” in Japan, in Australia, they fall under the “therapeutic goods” regulation category [7].

In pharmacology, bioactive peptides can be regarded as virtually ‘active principles’, with applications in human and animal medicine and a biological status transcending their nutritional value [8]. Needless to say, bioactive peptides must be well-tolerated and display only few (preferably no) minor side effects, including a lack of immunoreactivity [9]. We have recently become aware that foodstuffs, such as proteins, not only constitute nutrition sources by providing amino acids, the building blocks required for protein synthesis, but can also act on a variety of organs and tissues, modulating an array of physiological body functions [10]. According to these authors, despite the recent advantages in this field, the industrial applications of bioactive peptides are hindered by the lack of appropriate production methods and protocols; in fact, unless these limitations are overcome, their health applications will be seriously hampered.

Bioactive peptides can be generated by the action of proteolytic enzymes, such as proteases and peptidases, with the latter category mainly including proteins that catalyze the hydrolysis of substrates by an endo-splitting mechanism (as represented in Figure 1). Alternatively, bioactive peptides can be also produced by partial chemical hydrolysis in procedures involving incubation at different temperatures; in any case, the resulting compounds must be isolated and purified to fully evaluate their biological activities. The functions of some of these bioactive peptides were summarized by Chacrbarti and colleagues in 2018 [10]; these functions include “antihypertensive, antimicrobial, antithrombotic, immunomodulatory, opioid, antioxidant, and mineral binding functions”. As of 2014, more than 1500 bioactive peptides had been described [11,12], originating from either endogenous (generated the body’s own proteases, including biopeptides acting on body polypeptides) or external sources (proteins either included in the diet or produced by the action of proteases, which can be encoded by the human genome or by microbial genes). The term ‘encrypted peptides’ is often used to describe a polypeptide concealed within a larger protein, hence requiring the action of a protease to release the peptide; these compounds, although displaying a great sequence variability, usually contain up to 20 amino acids [13], which generally include proline and similar residues and, interestingly, are generally resistant to further hydrolysis by peptidases.

Bioactive peptides can be encrypted in the amino acid sequence of exogenous proteins present in the diet, and as indicated by Sánchez and Vázquez (2017) [12], the most important sources of such peptides are caseins from bovine milk, including those present in cheese and additional dairy products; these active molecules (summarized in Table 1) are normally released by the action of microbial proteases, such as those synthesized by milk-fermenting bacteria. However, milk is not the only source of animal-derived bioactive peptides, because they can originate from blood [14], gelatin, by the combined action of proteases from *Bacillus subtilis* and *Bacillus amyloliquefaciens* [15], and bovine serum albumin [16], as well as from proteins present in eggs or fish; however, regardless of their origin, these molecules are always the result of controlled proteolysis.

Some exogenous bioactive peptides that display antioxidative properties can be obtained by proteolytic processing of proteins present in food originating from either vegetable or animal matter. Plant-derived peptides can be produced from soy [17], potato, and corn [18], while milk caseins, eggs and meat are good sources of bioactive peptides from animal origins that exhibit antioxidant activity [19].

Some exogenous biopeptides display clear antimicrobial activity and are collectively known as antimicrobial peptides (AMPs) [20]; they are short amino-acid chains produced by practically all living creatures that help both animals and plants combat microbial infections, including viral diseases.

Microbial action, mainly by the gastrointestinal microbiota, on food proteins can generate bioactive peptides, including food-derived compounds with anti-inflammatory activity, and is exerted through the inhibition of signaling components belonging to either the NF-κB or the MAPK pathways [21,22], as well as proteases produced by psychrotrophic bacteria, which are microorganisms that can grow at temperatures used for refrigeration and storage of foodstuffs [23]. It is currently clear that diseases, such as type 2 diabetes, can be successfully treated simply by consuming diets that generate these bioactive anti-inflammatory peptides [24].

Furthermore, certain bioactive peptides, generated in the gut by a concerted microbial degradation of food proteins, are “candidates for a novel strategy for reduction and control of neurodegenerative diseases” [25]; they can also be useful in the treatment of obesity by controlling the energy balance of the host, as well as reducing the expression of transcription factors responsible for inflammation [26]. Bioactive peptides with antiviral activity are gaining ground as the drugs of choice to control certain viruses [27,28].

Proteomics can play an important role in the identification and characterization of bioactive peptides; this new field of study, known as ‘nutriproteomics’, is currently gaining momentum because new techniques are becoming available for the rapid and precise identification of bioactive molecules (Figure 2) [29]. High-resolution mass spectrometry perhaps represents the leading method for this purpose because it is sufficiently sensitive to detect and identify different types of analytes present in complex mixtures, even when they are available at low concentrations [30,31]. The two main strategies, although not mutually exclusive, that are employed for these studies are (i) bioinformatics and (ii) empirics, with the first strategy based on the analysis of peptide sequences present in databases, assigning a putative role for the molecules based on their homology to known bioactive peptides [31,32]. Capriotti et al. and Guerrero et al., after an extensive peptidome analysis of human milk, were able to characterize as many as 700 endogenous peptides, originating from the lysis of 30 milk proteins, following a conserved, and specific, proteolytic pathway; in our opinion, this research article represents a state-of-the-art publication with regard to the implementation of bioinformatics in the search for novel bioactive peptides. The second, empirical approach is, thus far, the most widely used and, according to Capriotti and coworkers (2016) [31], comprises eight steps: (1) selection of an appropriate food protein source; (2) isolation of the proteins present; (3) release of peptide fragments by controlled proteolytic degradation with either endogenous or exogenous proteases; (4) preliminary bioactivity screening of selected peptides; (5) peptide purification; (6) reevaluation of the biological activity of the selected biopeptide; (7) peptide identification by mass spectrometry analysis; and (8) in vivo or in vitro validation of the biological activity displayed by the molecule. The separation of bioactive peptides that display similar molecular weights can be difficult and must be achieved using traditional biochemical procedures, such as gel exclusion or ion-exchange chromatography, solid phase extraction, and high-performance liquid chromatography. These scientific methods, although satisfactory at a small scale, such as in laboratory procedures, may not be suitable for optimizing for commercial production, which occasionally requires the development of new procedures. These methods may be found in Ulug and colleagues in 2021 [33].

The following sections will address the concepts outlined in this introduction, with in-depth analyses of the putative sources of a great variety of bioactive peptides focused on food-derived bioactive peptides with anti-allergic and anti-inflammatory properties, which are consistently produced by a specific and structured mechanism requiring proteolytic cleavage of either endogenous or exogenous proteins, with the latter mainly being provided by foodstuffs.

## 2. Bioactive Peptides with Anti-Allergic Properties

Food allergy is mainly an IgE immune-mediated disease that results from a hypersensitive immune response to food allergens [34]. Around 6–8% of children and 2–4% of adults are affected by food allergies. A total of 14 foods are considered by the European Food Safety Authority (EFSA) as allergens. The only efficient and confirmed treatment for food allergies at present is to ingest a diet free of the causative food and its byproducts. In addition to personal vulnerability, the ability of allergen proteins or their associated food matrix constituents to stimulate Th2 pathways over Th1 immunity is believed to be key to determining their ability to induce an allergic response [35].

The most favorable strategies at this time, which are assessed against IgE-mediated allergic diseases, include the opportunity to use proteomics for the characterization of epitopes to develop new food allergy vaccines [36,37] and the use of immunomodulating peptides to stimulate oral tolerance toward food allergens or prevent allergic sensitization [38]. Accordingly, several potential food sources of anti-allergens were obtained based on bioactive peptides.

Table 2 summarizes the list of food-derived bioactive peptides with anti-allergic properties that have been published.

Eggs have been recognized as a valuable source for the provision of biologically bioactive peptides. The two main egg allergens are ovalbumin (OVA) and ovomucoid (OVM). Yang et al., 2009 [39] characterized the immunomodulatory results of egg white enzymatic hydrolysates using a BALB/c mouse model of egg allergy against OVA. Animals were orally sensitized to egg white and then treated with enzymatic hydrolysate. The characterization of peptides was achieved using MALDI-TOF-MS with online LC-MS/MS analytical equipment. The bioactive peptides (AMVYLGAKDSTRTQ, SWVESQTNGIIRNVL and AAHAEINEAGREVVG) demonstrated by ELISA significantly reduced both serum histamine and specific IgE titers in enzymatic hydrolysate-fed mice, complemented by the suppression of both IL-4 and IFN-γ production in spleen cell cultures. The occurrence of the immunodominant biopeptides was suggested to be in charge for the detected immunomodulatory results.

Similarly, Rupa et al., 2015 [40] investigated oral immunotherapy with the immunodominant biopeptide (DNKTYGNKSNFSNAV) for the major egg white allergen ovomucoid (OVM) in a BALB/c animal model of egg allergy. The biopeptide-treated mice had fewer histamine and IgG1 and more IgG2-related antibodies, representing a bias toward the type-1 reaction. Additionally, an important increase in the percentage of Tregs in the epitope-treated group was detected. The obtained results validated the prospective use of the OVM (DNKTYGNKSNFSNAV) peptide as an immunoregulator for egg allergy.

Regarding cow’s milk allergy, several epitopes of β-lactoglobulin (BLG) were utilized in presensitized BALB/c animals [41]. Among them, the two dominant epitopes (AQKKIIAEKTKIPAVFKIDALN and ALKALPMHIRLSFNP) were used to gavage BLG-sensitized mice. In contrast to un-treated control mice, both peptide-treated groups had inferior hypersensitivity scores with unchanged rectal temperatures and similar BLG-specific serum immunoglobulin levels and Treg cell populations. Symptoms of allergic reactions were relieved in peptide-treated mice.

Casein peptide (HAQ) from cow’s milk showed anti-allergenic effects in vitro and in vivo [42]. The obtained results showed that the casein HAQ peptide produced through hydrolysis with digestive and other enzymes can repress the antigen-induced degranulation of RBL-2H3 cells. Moreover, using a murine type-1 allergy model of C3H/HeJ mice, the continuous dispensation of HAQ peptide suppressed mild allergic symptoms in an animal model of type-1 allergy [42].

Fermented food compounds are also important immunoregulatory products that have anti-allergy properties due to the production of lactic acid bacteria [52]. Previous studies have showed that the consumption of fermented milk products (i.e., yogurt) containing lactic acid bacteria, such as *Lactobacillus* and *Bifidobacterium,* can enhance the production of Type I and Type II interferons at the systemic level [53]. In mouse models, lactic acid bacteria have been shown to stimulate interferon release and to decrease allergen-stimulated production of IL-4 and IL-5 [54]. Thus, some well-defined strains of immunoregulatory lactic acid bacteria can downregulate a Th2 allergic phenotype [54].

Different bioactive peptides were prepared using a combinatorial chemistry approach (LSYLLWRSRLP, LVAHVGAGGVL, RVSSCRGRNHIV, ETIGARWVRIE, TDGVTYTNDCL, RVVRYDADFWI, GFWCRRSGLVGV) denoted as Protein A Mimetic (PAM) and they can bind to immunoglobulins of different types and species and reduce the release of histamine, PCA and inflammatory cytokines (TNF-α, IL-1β and IL-6) [43]. The obtained results showed that these peptides were active in the studied models, with effective doses below the toxicity level; hence, the abovementioned molecule is a promising candidate for the development of a new class of anti-allergic drugs through the biopanning of pooled sera of patients with different allergies [44].

Maria Rossi prepared and polymerized a tripeptide (RTY) and polymerized using chemical synthesis. It was found that this tripeptide could efficiently inhibit mast cell degranulation and the release of β-hexosaminidase from rat basophils and decrease allergic reactions [43,45].

Cashew-nut protein hydrolysates with antiallergic activity were formulated from cashew nuts through protease action with five different enzymes (alcalase, protamex, neutrase, papain, and bromelin) [55]. The fraction obtained with alcalase and fractionated thought ultrafiltration (<3 kDa) showed the highest hyaluronidase inhibitory rate (90.57%) and could be used as a possible source of antiallergic peptides for the development of remedies to treat allergic reactions to cashew nuts.

Regarding marine products, antiallergic peptides from the enzymatic hydrolysate of *Spirulina maxima* peptides P1 (LDAVNR) and P2 (MMLDF) were identified by mass spectrometry techniques [46,47]. In addition, peptides purified from the enzymatic hydrolysate of *Spirulina maxima* [48] showed that ADSDGK could reduce the incidence of an allergic reaction by 75% at a dosage of 5.0 mg/kg in a passive allergy test in mice and had strong anti-allergic activity.

Additionally, the peptide (PFNQGTFAS) from the abalone intestine gastrointestinal digests [49] was indicated to be a favorable candidate of anti-allergic therapeutics for mollusks.

Furthermore, Atlantic salmon byproduct hydrolysates with pepsine were purified by Sephadex G-15 gel permeation chromatography, HPLC, and LC, and the novel peptides were identified through LC/MS/MS [50]. Among the RP-HPLC fractions, the C6 fraction showed the strongest anti-allergic activity (89.40%; *p* < 0.05) at 1 mg/mL. Then, the new peptide identified as TPEVHIAVDKF was proven to exert anti-allergic activity after synthesis by inhibiting the release of β-hexosaminidase in IgE-mediated RBL-2H3 cell degranulation at an IC50 value of 1.39 mg/mL. Therefore, Atlantic salmon byproducts can be a probable source of novel ingredients in food and pharmaceuticals for food allergy controlling [50,51,56].

All these bioactive peptides can be a possible source of novel drugs and ingredients in food and pharmaceuticals for food allergy management.

## 3. Bioactive Peptides with Anti-Inflammatory Properties

Inflammation is part of the biological response by which the immune system defends the body from harmful inducers, including pathogens, cell injury, toxic substances, or irritation [57]. A general inflammatory process involves four phases: induction of inflammation, detection of inflammation by inflammatory sensors, production of inflammatory mediators, and target tissues that are modulated by the inflammatory mediators [58]. The sensors, mediators, and target tissues may be different depending on the type of inflammatory stressor involved; thus, this mitigation process could better contribute to restoring tissue homeostasis and mitigating acute inflammation. Therefore, inflammation is a key process to counteract injury or infection and maintain body health [57]. However, when inflammation is uncontrolled and persists for a long time, it may become chronic, contributing to the progression of an important variety of diseases, such as atherosclerosis, type-2 diabetes, skin diseases, and several aging-related diseases [21,57].

Nonsteroidal anti-inflammatory drugs (NSAIDs), such as ibuprofen and aspirin, are widely used to manage inflammation. However, the presence of well-known side effects, such as gastric bleeding, ulceration or renal dysfunction, complicates the long-term use of NSAIDs for a large part of the population [21]. Given the side effects derived from the long-term use of NSAIDs, food-derived bioactive peptides from both plant and animal sources have been investigated for their anti-inflammatory properties to provide safer alternatives. The anti-inflammatory effect of food bioactive peptides has been extensively tested in vitro in cultured mammalian cell systems and, to a lesser extent, in vivo in various animal models of human diseases. Peptides from different food sources have been described to have a neutralizing effect on the inflammatory process. A compilation of peptides with well-known structures and exhibiting anti-inflammatory effects either in cell or animal models are summarized in Table 3.

Peptides obtained from fish and shellfish proteins have shown relevant anti-inflammatory effects. Gao and collaborators [59] identified peptide sequences with potential anti-inflammatory activity derived from sturgeon muscle protein in the lipopolysaccharide (LPS)-induced RAW264.7 cell inflammatory model. Fourteen novel peptides were identified by LC-MS/MS by Q Exactive Hybrid Quadrupole-Orbitrap mass spectrometry, and three peptides were synthesized (KIWHHTF, VHYAGTVDY, and HLDDALRGQE). These synthetic replicates were able to decrease the release of inflammatory mediators and inflammatory cytokines (NO, IL-6, and IL-1β), while significantly increasing superoxide dismutase (SOD) activity in the cell model. These peptides also downregulated the phosphorylation levels of MAPKs, indicating that such peptides exerted anti-inflammatory effects by inhibiting the MAPK pathway.

Salmon byproducts have been reported to be important sources of fish peptides with anti-inflammatory properties. An enzymatic hydrolysate of *Salmon salar* skin exhibited strong anti-inflammatory activity [60]. This hydrolysate was further separated and identified by ultrafiltration followed by LC-MS/MS analysis. The peptides APD, QA, KA, and WG reduced the secretion of NO, IL-6, IL-1β, and TNF-α in inflammatory macrophages. Among them, peptide QA showed the highest anti-inflammatory activity. A relevant characteristic of most of the identified anti-inflammatory peptides was their stability against digestion [60]. An anti-inflammatory peptide from salmon pectoral fin protein hydrolysate generated by pepsin hydrolysis was purified using Sephadex G-25 gel permeation chromatography [61]. The purified anti-inflammatory peptide was identified by LC-MS/MS analysis as the tripeptide PAY. This peptide was able to inhibit the production of nitric oxide (NO) and prostaglandin E2 (PGE2) in RAW264.7 cells under lipopolysaccharide treatment. Additionally, the peptide PAY significantly reduced the production of the proinflammatory cytokines tumor necrosis factor-α, interleukin-6 and interleukin-1β [61].

Protein hydrolysates from tuna cooking juice also provided several peptides with anti-inflammatory properties that could to induce secretary and cellular responses in murine peritoneal macrophages from RAW264.7 cells [62]. The peptide with higher activity was identified as PRRTRMMNGGR by an LC-QToF mass spectrometer coupled with an electrospray ionization (ESI) source. This peptide inhibited the secretion of inflammatory cytokines (TNF-α, IFN-γ, and IL-2) released by LPS-stimulated RAW 264.7 macrophages.

A bioactive peptide from a peptic hydrolysate from *Meretrix* clams was also characterized as an anti-inflammatory agent [63]. The peptide sequence was identified as NPAQDC by liquid chromatography-tandem mass spectrometry (LC-MS/MS). Hexapeptide was able to significantly reduce cyclooxygenase (COX)-2 activation and both proinflammatory cytokine and nitric oxide (NO) production in RAW264.7 macrophage cells.

Milk is another food source of bioactive peptides. In previous studies, the milk casein-derived peptide QEPV was isolated from fermented milk [65]. The obtained results showed that QEPVL significantly activated lymphocytes both in vitro and in vivo. Lymphocytes have an important function in mediating specific immune responses against pathogens by secreting antibodies or cytokines. This peptide inhibited LPS-induced inflammation by regulating nitric oxide release and the production of cytokines IL-4, IL-10, IFN-γ, and TNF-α in vivo. The artificial gastrointestinal digestion of QEPVL yielded the digestion product QEPV, which was identified by ultra-performance liquid chromatography/quadrupole time-of-flight mass spectrometry (UPLC-Q-TOF-MS) analysis. QEPV exhibited bioactivities similar to those of QEPVL in vitro. Overall, QEPVL has significant immunomodulatory effects on lymphocytes and contributes to the treatment of inflammation through the oral route as a functional food ingredient. VPP and IPP are other milk-casein-derived peptides exhibiting anti-inflammatory properties. These peptides have exhibited anti-inflammatory activities by inhibiting the activation of the NF-kB pathway and the reduction of adipokine levels in murine preadipocytes [69]. In another study, both peptides were also able to attenuate atherosclerosis development in apolipoprotein E-deficient mice by reducing the expression of the proinflammatory cytokines IL-6 and IL-1b and the NF-kB-related genes CD40, LCK, PIK3CG, IL1B, and MAP2K7 [68].

In vitro digested human milk and in vivo pooled intestinal samples from infants fed human milk also provided fractions and peptides with inflammatory modulating activities [64]. Peptides extracted from the pooled in vivo intestinal samples attenuated both TNF-α and IL-8 secretion in LPS-treated human immune THP-1 macrophages. There were 266 and 418 peptides identified by LC-MS/MS–based peptidomics in both in vitro and in vivo samples. Two fractions were found to exhibit proinflammatory activity, while three fractions showed a significant anti-inflammatory activity. Mass spectrometry analysis was able to identify 13 peptides in all fractions with anti-inflammatory activity, and 38 peptides in all fractions with proinflammatory activity.

Several plant peptides with anti-inflammatory properties have also been identified by proteomic tools. For instance, anti-inflammatory activity was found in peptide extracts extracted from gastrointestinal digests of bean milk and yogurt in both Caco-2 mono- and Caco-2/EA.hy926 coculture cell models [66]. Three peptides, including γ-glutamyl-S-methylcysteine (γ-E-S-(Me)C), γ-glutamylleucine (γ-EL), and LLV, were found to be transported across the Caco-2 cell monolayer and detected by liquid chromatography-tandem mass spectrometry. A strong anti-inflammatory effect was observed in basolateral EA.hy926 cells (coculture model) as a result of the inhibition of tumor necrosis factor α-induced pro-inflammatory mediators of the nuclear factor κB (NF-kB) and mitogen-activated protein kinase (MAPK) signal cascades. These results suggested that these peptides can be absorbed and could possibly have a systemic inhibition of inflammatory responses in vascular endothelial cells, indicating potential preventive effects on vascular diseases.

A peptide sequenced as KQSESHFVDAQPEQQQR was obtained from extruded adzuki bean protein by simulated gastrointestinal digestion by ultrafiltration and pre-HPLC separation and identification by UPLC-MS/MS analysis [67]. This peptide was shown to exert significant anti-inflammatory effects in LPS-treated RAW 264.7 macrophages by reducing the production of IL-1, IL-6, TNF-α, and MCP-1. The involved signaling pathways were significantly related to the NF-κB pathway.

Another work reported the anti-inflammatory properties of a natural peptide network (NPN) from rice, describing that the NPN was more intense as an anti-inflammatory than DHA regarding TNF-α secretion in THP-1 macrophages [70]. The NPN was further studied by LC-tandem mass spectrometry (LC-MS/MS), and this permitted the identification of seven peptides with remarkable anti-inflammatory activity: among these, NGVLRPGQL and SEEGYYGEQQQQPGMTSR provided the largest reductions in TNF-α secretion in THP-1 macrophages [70]. In another work, these authors described that when NPN isolated from rice was incorporated as a supplement in an elderly cohort for 12 weeks, it was able to decrease the concentration of cytokines and CRP inflammatory biomarkers. Nonetheless, the differences found between the treatment and placebo cohorts were only significant in the area under the curve (AUC) of TNF-α.

Other works described the anti-inflammatory activities of extracts isolated from other plant sources. In this sense, peptides LDAVNR and MMLDF isolated from spirulina inhibited IL-8 production by EA.hy926 endothelial cells [71]. A list of anti-inflammatory compounds of plant origin, including 39 bioactive peptides, has recently been published [60]. This list compiles peptides isolated from sources such as hempseed, millet bran, brown rice, walnut, chia seed, green tea, bee pollen, rice bran, corn, adzuki bean, lupin, sunflower and wheat germ. Interestingly, the amino acid composition of such peptides, such as the presence of leucine and isoleucine, has been reported to reduce the damage caused by inflammatory mediators by inhibiting the kinase phosphorylation pathway [72]. Peptide hydrophobicity has also been reported to be connected to the intensity of the anti-inflammatory activity [73]. Thus, highly hydrophobic peptides can cause the reversion of cell membrane charge, thus avoiding the inflammatory response caused by LPS. The presence of hydrophobic amino acids at the N-terminus has also been linked to more intense anti-inflammatory activities [60]. In contrast, the presence of positively charged amino acids, such as lysine, arginine and histidine, has been described to increase the intensity of anti-inflammatory activity as a consequence of different molecular mechanisms [62,74,75]. Likewise, the presence of certain amino acids, such as glycine and glutamine, has also been linked to more intense anti-inflammatory activity: these are the cases of the presence of glycine in four peptides from sunflower [76] and the presence of glutamine in five peptides from maize [77].

## 4. Bioactive Peptides by Proteomics

Proteomic techniques for peptide identification and the characterization of bioactive peptides include classical bottom-up methods, such as two-dimensional gel electrophoresis and advanced top-down methods based on mass spectrometry (MS) for food “protein fingerprinting” and “peptide mass fingerprinting”. Moreover, appropriate computer programming is usually also necessary for the final identification of bioactive peptides [78].

### 4.1. Recent Proteomic Approaches in the Identification and Quantification of Bioactive Peptides

For the identification and quantification of bioactive peptides, previous protein/peptide separation is necessary. Electrophoretic and chromatographic technologies, separately or in combination, both offline and online, have been developed to achieve protein/peptide separation [79]. Polyacrylamide gel electrophoresis based on two-dimensional gel electrophoresis (2-DE) technology is one of the most popular and effective methods. Liquid phase separation, such as capillary electrophoresis (CE) or liquid chromatography (LC), is another separation method used for protein and peptide separation [80]. LC and CE utilization is increasing due to its advantages such as high sensitivity, superior dynamic range, easy automatization, speed and versatility compared to 2-DE [81]. The combination of conventional membrane filtration with electrophoresis, and even the use of electromembrane filtration (EMF), has also been considered as a method for the isolation of bioactive peptides [82].

X-ray crystallography, mass spectrometry (MS) and nuclear magnetic resonance (NMR) methods, in combination with different software analyses, can be used to study the 3D structure and functional properties of food-derived bioactive peptides [82]. Additionally, ELISA, as an immunochemical technique, can be used for the detection and determination of very small quantities of peptides in food because it is more sensitive than HPLC and as specific as the radioimmune assay [17,83].

The preferred analytical tools employed for the isolation and determination of bioactive peptides are chromatography and mass spectrometry techniques, such as high-pressure liquid chromatography (HPLC) [78]. HPLC techniques use several detectors for the qualitative and quantitative determinations of bioactive peptides such as UV/VIS, photodiode, and fluorescence (based on absorbance or fluorescence values). In addition, adulteration of peptide-containing products of animal origin and plant products can be detected by chromatographic techniques [84,85]. On the other hand, in the MS proteomic analyses for the quantification and identification of bioactive peptides, several ionization methods are employed, including electrospray ionization (ESI) or matrix-assisted laser desorption ionization (MALDI). Quadrupole analyzer (Q), ion trap (IT or QIT), and time of flight (TOF) have also been employed for mass identification [78].

Liquid chromatography (LC) is usually combined with mass spectrometry (MS) (LC-MS). Thus, liquid chromatography followed by tandem mass spectrometric detection (LC–MS/MS) has become the standard method for the characterization of peptide sequences [86]. Relatedly, different methods have been developed in recent years for the analysis of active biopeptides, such as liquid chromatography-electrospray ionization-tandem mass spectrometry (LC-ESI-MS/MS) with low-resolution quadrupole mass analyzers [87,88,89] and a high-resolution orbitrap system [88] or off-line systems, including ESI-MS/MS used without LC [87] and fast atom bombardment [90]. Intact milk proteins and their derivatives, such as fermented dairy products, hydrolysates from simulated gastrointestinal digestion of milk protein fractions, or hypoallergenic infant milk formulas, were analyzed using these techniques [91,92,93,94].

Mass spectrometry is also combined with capillary electrophoresis (CE-MS) and is typically utilized to separate and characterize a wide range of compounds, particularly polar and charged molecules, such as peptides and proteins [94,95,96,97,98,99,100,101,102,103]. CE has been applied in several studies, providing information about proteolytic profiles, from milk and cheese ripening and studies of alterations in soybean proteins [104,105,106,107]. Peptide qualities, safety, functionality, and food processing have also been published [108]. The principal advantage of CE is the possibility of equipment miniaturization and the low consumption of reagents [109,110]. CE-MS also provides a rapid, efficient, and sensitive method analyzing f food allergens [111].

Considering the different properties of the mass spectrometers, the analyzer can be used individually or in combination as a Q-TOF, TOF/TOF, ion trap (IT), orbitrap, and Fourier transform ion cyclotron resonance analyzer (FT-ICR). Due to the potentially unlimited mass acquisition range in a single-run, high-resolution, mass-accuracy, reducing bench space and the high acquisition rates of the current types of MS instruments; they have been used for the identification and characterization of bioactive peptide sequences [94,112]. Thus, bioactive peptides were analyzed using accurate molecular weight data obtained by CE-TOF-MS [94]. Moreover, other analyzers, such as Q, IT, TOF, and FT-ICR, are coupled with CE and generally use ESI as the ionization source. An example includes the CE-ESI-TOF-MS online method and CE-MALDI-TOF/TOF-MS offline method [96]. Recently, Carrera et al., 2019 used a high-resolution orbitrap instrument to identify and characterize bioactive peptides from Jumbo squid skin byproducts by shotgun proteomics.

There are some challenges associated with MS peptide identification that should be considered, including the high mass accuracy, number of copies in a complex mixture, peptide length, collision energy used for fragmentation, peptide retention time, and computer-based validation [113]. Amino acid sequences can be identified in an extensive chromatographic system containing a diode array detector (DAD), for example, HPLC-DAD-ESI-MS and through tandem MS or mass selection/mass separation experiments [91,114]. Tandem mass spectrometry is a more accurate method used in proteomic studies to determine the amino acid sequence of peptides.

In addition, MS is used for the determination of peptide mass fingerprinting. Matrix-assisted laser desorption/ionization time of flight (MALDI-TOF) mass spectrometric analysis is the basis for obtaining the peptide profiles of protein hydrolysates or even fractions [86]. In this methodology, protein extraction is digested, often using trypsin, and hydrolysates received by the enzymatic reaction can be studied using a MALDI-TOF mass spectrometer to determine the peptide structure and mass verification. The “peptide mass fingerprinting” technique determines these peptides using necessary computer programming [78]. The MALDI technique is fast because sample measurement does not exceed a few nanoseconds; however, sample preparation is tedious, and the separation of a complex matrix is also a problem [115]. The development of research combining two-dimensional electrophoresis techniques with MALDI-TOF spectrometry has been performed for many years; in this analysis, the mixtures of soluble proteins are initially separated by bidirectional electrophoresis. Gel spots are cut out of the gel and analyzed by MALDI-TOF, and the intensity of the mass spectra depends on the proportion and type of the obtained ions. Ayala-Niño and collaborators have identified the sequence of bioactive peptides from amaranth seed proteins [116]. MALDI-TOF was also used for lunasin identification in several amaranth seed [117] and soybean varieties. Moreover, the effect of environmental factors on the lunasin concentration [118] has also been determined. The use of MS/MS spectra for peptide sequencing is advised when bioactive food peptides result from the enzymatic hydrolysis of proteins [119,120]. Researchers have also designed high-performance liquid chromatography with evaporative light scattering detection (HPLC-ELSD) as an investigation tool developed for purification and quantitative measurements [86].

In recent years, many studies have been performed for the analysis and identification of different peptides using previously described techniques. As the efficiency and speed of analysis have become essential for bioanalysis, ultra-high-performance liquid chromatography-tandem mass spectrometry (UHPLC-MS/MS) analysis and rapid-resolution liquid chromatography-tandem mass spectrometry (RRLC-MS) provide important advantages in this area [86]. Thus, some studies have been performed to isolate bioactive peptides adquired by enzymatic hydrolysis of food proteins by reverse-phase high-performance liquid chromatography (RP-HPLC) [82,120,121,122]. Moreover, the addition and combination of MS/MS with RP-HPLC has been employed in many studies [123], such as bioactive milk protein peptides obtained in the hydrolysis process of the human gastrointestinal tract. Ultrafiltration (UF) and reversed-phase high-performance liquid chromatography (RP-HPLC) were used to purify buffalo casein hydrolysates produced by trypsin and alcalase, and four antioxidant peptides were identified by LC MS/MS [124,125]. Moreover, IR-MALDESI MS was used to determine bioactive peptides in fermented cucumber. Peptides were identified by IR-MALDESI MS/MS and quantified by LC-MS/MS [126]. Furthermore, conventional mass spectrometry-based proteomics can be complemented using a data-independent acquisition method such as sequential window acquisition of all theoretical mass spectra (SWATH-MS). These methods have been used to classify and quantify peptides derived from trypsinogen, ferritin, and uncharacterized proteins in the enzymatic hydrolysis of porcine liver [127]. In another study, nano-LC-MS/MS was used for tilapia skin collagen protein sequence identification [86]. Another approach showed that goat milk fermentation processing increased the amount of protein in goat milk samples. The fractions were separated by SDS-PAGE, and the resulting bioactive peptide fractions were analyzed and identified by LC MS/MS [128].

In addition, other in vitro studies of the health-promoting effects of food-derived bioactive peptides with identified sequences were performed. These examples include antioxidant peptides from cottonseed [129], jackfruit [130] and shrimp [131] that were identified by UPLC-MS/MS, UPLC-Q-TOF-MS/MS, and UPLC-MS, respectively. Moreover, bioactive peptides from the sarcoplasmic fish proteome of 15 different fish species were determined by LC-orbitrap analysis [132].

Furthermore, anti-inflammatory peptides from *Baiju vinasse* [133], sturgeon muscle [134], and amaranth [135] were determined by mass spectrometry techniques. Anti-inflammatory peptides were obtained from chickpea [136], pine needles [137], buffalo casein [138] and *Satureja khuzistanica* leaves [139], and identified by MicroLC-ESI-Q-TOF-MS/MS, MALDI-TOF-MS, LC-MS/MS and nano-HPLC-ESI-MS/MS, respectively. Anti-allergic peptides from Atlantic salmon and *Spirulina maxima* [46] were identified by mass spectrometry techniques. Atlantic salmon byproduct hydrolysates have been purified by Sephadex G-15 gel permeation chromatography, HPLC, and LC, and the novel peptides were identified through LC/MS/MS [50,56].

Among in vitro studies of the health-promoting effects of food-derived bioactive peptides, antihypertensive peptides were identified from porcine blood [140], *M. longissimus* beef [141], and wheat germ [142] by MS/MS. The identification of ACE-inhibitory peptides produced by enzymatic hydrolysis of cuttlefish wastewater was presented in [143]. In addition, hypocholesterolemic peptides from *S. platensis* and antidiabetic peptides from camel milk [144], camel whey [145], walnut [146], cumin seed [147] and chickpea [148] have been reported.

In addition, other in vivo studies of the health-promoting effects of food-derived bioactive peptides with identified sequences were performed. These examples include antihypertensive peptides from wine lees identified by nano-HPLC (orbitrap) MS/MS [149] and anti-adipogenic peptides from defatted mealworm identified by MALDI-TOF-MS [150].

Macroalgae, also known as seaweed, constitute ancillary putative sources of bioactive peptides; these aquatic plants, in addition to providing suitable habitats for marine animals, have an economic potential estimated to be worth USD 6.4 billion [151]. Although conventional proteomics can be implemented in macroalgae using 2-DE, MALDI-TOF MS, MALDI-TOF/TOF MS, or nano-LC-MS/MS [72,152,153,154] to detect potentially bioactive peptides, in some cases, transcriptomics are a better choice; this applies, in particular, when the algae are subjected to abiotic stress, which can be due to lead or cadmium contamination, as well as unfavorable temperatures [155,156,157,158,159,160].

Finally, a good strategy and procedure for shotgun proteomics and protein-based bioinformatics approaches for the characterization of food-derived bioactive peptides was presented by Carrera et al., 2021 [161].

### 4.2. In Silico Approaches for Bioactive Peptides

Sanger [162] sequenced the first protein in 1952; millions of proteins have been characterized, and their sequences have been deposited in searchable online databases such as UniProtKB (UniProt Consortium), the NCBI Protein database (National Center for Biotechnology Information), and the Protein Data Bank (Research Collaboratory for Structural Bioinformatics) [163].

Bioinformatics, together with proteomics, provide for the study of the structure–activity relationships and sequence information of potential bioactive peptides (Figure 3) [161]. Bioinformatics and in silico studies may predict bioactive sequences encrypted in food proteins. Furthermore, together, bioinformatics and proteomics approaches are effective techniques for predicting, profiling, and characterizing bioactive protein hydrolysates and peptides from food. Proteomics has been implemented to diverse fields of research, including food science, for identifying and quantifying peptides with biofunctional relevance. For the analysis of mass spectrometric data, bioinformatic software uses algorithms to process and validate peptide obtained data (mass of peptide fragments) and contrast them with databases (e.g., Universal Protein Resource (UniProt Knowledgebase) protein sequence collections, National Center for Biotechnology Information (NCBI) protein sequence collections, and proteomic databases of the European Bioinformatics Institute (EBI) and OWL) [78]. This analysis may identify proteins by comparing molecular weights and amino acid sequences of the obtained peptides [164,165,166], enabling the identification of peptides after translational modifications, during the process of digestion, and de novo sequencing. The identification of posttranslational modifications (PTMs) would be impossible without the use of updated databases and computer softwares. This is due to the very cumbersome origin of the results in the analyses of these samples [78].

Bioinformatics have the ability to predict physicochemical and biological (e.g., antimicrobial, enzyme inhibition, allergenicity, toxicity) properties for the design of novel bioactive peptide sequences, which is useful for functional food or pharmaceutical applications [163]. Chemometrics provides useful information from multidimensional measurement data using statistics and mathematics [168]. The employement of chemometric and bioinformatics methods decrease cost and time during research [78]. Bioactive peptide performance can be predicted by QSAR and in silico bioinformatics-based methods [169]. Information relating to the chemical structure is compared and searched by QSAR to correlate the biological activity with the structural differences of the compounds [170]. In food science, in silico analysis is used for screening and searching for potential sources of bioactive peptides in different databases, such as BIOPEP (focusing mainly on peptides of food origin) [164], SwePep, PeptideCutter, POPS, NeuroPred BioPD, EROP-Moscow, and PepBank [78,168]. The simulation of protein hydrolysis with the elucidation of cleavage specificities of different proteases is considered in the generation of peptides from a protein [82]. The most commonly used bioinformatics tools for in silico protein digestion are BIOPEP BEnzyme action and ExPASy PeptideCutter [163,168,171]. Following digestion or peptide sequencing, the peptides can be in silico characterized for their physicochemical properties (such as molecular weight, theoretical pI, aliphatic index, average hydropathicity, etc.) and biological properties (such as different bioactivities (ACE-inhibitory, antimicrobial, antiallergenic, antioxidative), toxicity and potential allergenicity). Moreover, there are bioinformatic tools for the characterization of sensory properties such as the presence of sweet, bitter, umami, and other taste-evoking peptide sequences [163].

Molecular docking studies are other bioinformatic tools for peptide bioactivity prediction and are often employed for the design and screening of bioactive molecules and the selection of the drug candidates [172]. This technique has been employed in several studies for the evaluation of the select peptides [163]. For example, Chamata et al. used molecular docking [173] to describe the relationship between the structure and angiotensin-converting enzyme inhibitory activity of peptide sequences present in whey/milk protein hydrolysates. Table 4 shows that the principal in silico methods include databases, online tools, and software.

Moreover, other food-derived bioactive peptides have been analyzed by bioinformatics in cereal proteins [174,175], soybean proteins [176], egg proteins [177], milk proteins [178], cheese [179] and meat proteins [180,181,182]. *Chlorella sorokiniana* proteins are a source of bioactive peptides [183], and an in silico analysis carried out by BIOPEP has shown multiple dipeptydil peptidase IV inhibitors, glucose uptake stimulants, antioxidants, and regulating, antiamnestic and antithrombotic peptides [184]. In another study, tilapia proteins were determined by proteomic techniques and analyzed using a BIOPEP peptide cutter, revealing the best proteases for obtaining the highest number of potential angiotensin converting enzyme (ACE)-inhibitory peptides and antithrombotic and anti-amnesic activities from tilapia collagens [185]. In a previously described study, Chen et al. used nano-LC-MS/MS for tilapia skin collagen protein sequence identification following SAR analysis of these peptides, which had a great effect on ACEi activity. The AFTpin platform has been used to predict the ACEi activity of peptides [186]. In addition, bioactive peptides released from giant grouper (*Epinephelus lanceolatus*) were characterized by SDS-PAGE, in-gel digestion, mass spectrometry and online Mascot database analysis and by in silico analysis using the BLAST and BIOPEP-UWM databases for predicting potential bioactivities [187].

## 5. Concluding Remarks and Future Trends

It has been shown how several bioactive peptides from foodstuffs show functional activities, such as anti-allergic and anti-inflammatory effects [142]. Peptides with anti-inflammatory properties from different food sources can be used to provide safer alternatives to NSAIDs. Moreover, the use of proteomics for the determination of epitopes is a promising strategy against IgE-mediated allergic diseases, with immunomodulating peptides being used to induce oral tolerance and the design of new food allergy vaccines [36,37].

Microbial fermentation remains an inexpensive strategy to produce bioactive peptides in foods. Furthermore, the design and employment of genetically modified strains will become important because they release large amounts of proteolytic enzymes to hydrolyze food proteins. Additionally, the market of food-derived bioactive peptides will increase in the near future, and they will be sold as nutraceuticals [2].

Due to the complex nature of these compounds, their analysis requires the use of advanced techniques. Indeed, food proteomics technologies, such as the mentioned separation methods (such as chromatography and electrophoresis), mass spectrometer techniques, biological activity tests and bioinformatics (databases and programs processing peptide sequences, as well as predicting structure and physicochemical properties) appear to be interesting, necessary and promising sources of information to analyze bioactive peptides and enable rapid progress in this area [88,166]. Recent advances in high-resolution MS have allowed for faster and more exhaustive proteome and peptide characterization.

Functional foods and nutraceuticals can be designed by adding biologically active peptides into their formulation [125]. Therefore, only a few of the described bioactive peptides are available for human consumption at present, most of them of fish and milk protein origin. Further exploration of the technofunctional features of isolates and hydrolysates, alone and in combination with novel food products, is required [188]. Additionally, further research and new techniques (such as encapsulation) are needed to enrich active peptides from food proteins and to stabilize the chemical structures and biological activity of peptides in different food matrices [142].

## Figures and Tables

**Figure 1 nutrients-14-04400-f001:**
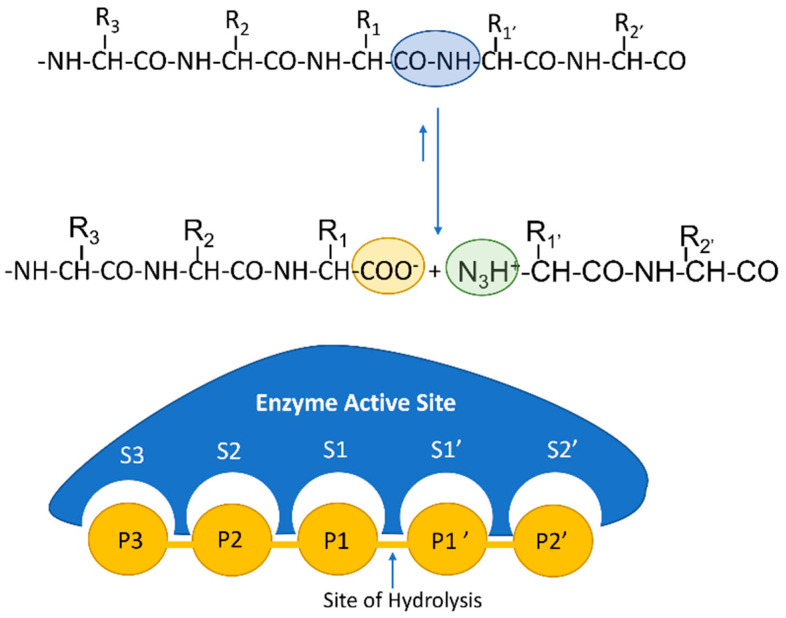
Graphic representation of the proteolytic degradation (hydrolysis) of a protein by the action of a protease. The enzymatic molecule contains an active site that cleaves the peptidic bond between two amino acids and involves several subsites (represented as semicircles and labeled S1–3 and S1’–2’) that allow for interaction between the enzyme and a specific sequence within the substrate, thus facilitating recognition of the peptidic bond that is to be cleaved. Hydrolysis results in two new peptides (diagram based on the publication by Baker and Numata, 2013).

**Figure 2 nutrients-14-04400-f002:**
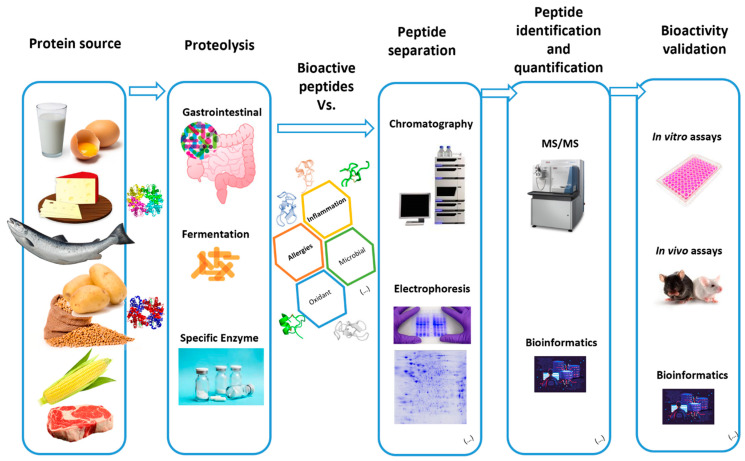
General schematic diagram of bioactive peptide production from foodstuffs and the sequential methods and analysis for its identification and characterization.

**Figure 3 nutrients-14-04400-f003:**
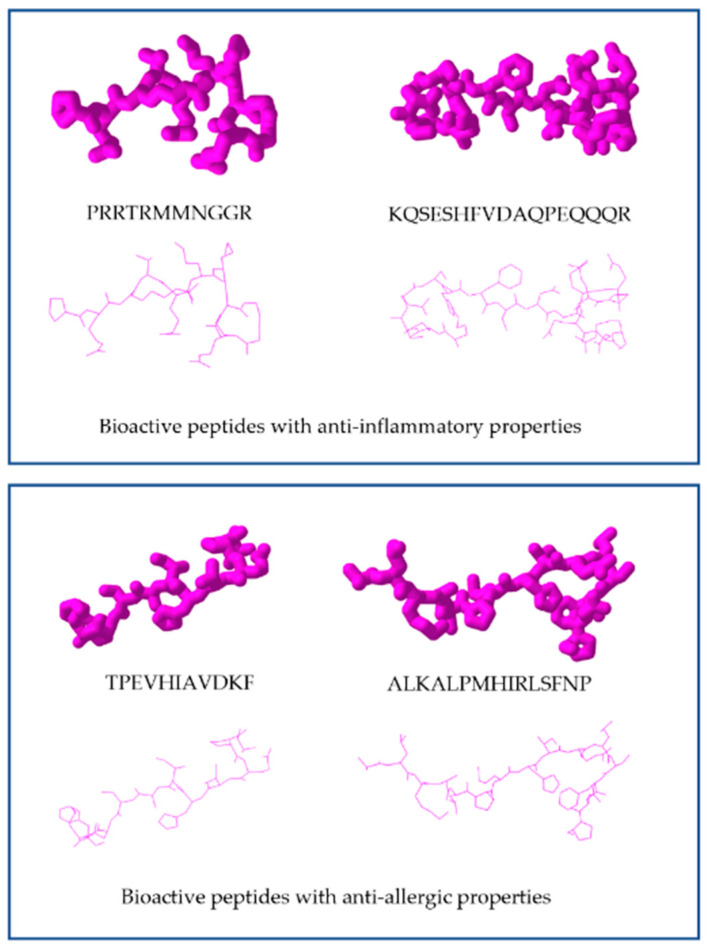
Representation of the secondary structure of different peptides that display anti-allergic and anti-inflammatory properties. Peptide structures were generated by the PEPstrMOD server [167] and visualized in Icn3d viewer (https://www.ncbi.nlm.nih.gov/Structure/icn3d/full.html).

**Table 1 nutrients-14-04400-t001:** Relevant bioactive peptides obtained from bovine milk caseins by the action of microbial proteases ^a^.

Source	Peptide	Activity
*Lactobacillus rhamnosus*, plus digestion with pepsin	DKIHPFQEPVL	ACE ^b^ inhibitor
*Lactobacillus helveticus*	VPPIPP	ACE inhibitor
*Lactobacillus* GG plus pepsin and trypsin	YPFPAVPYPQRTTMPLW	Opioid, ACE inhibitor, immune-stimulator
*Lactobacillus delbrueckii* subsp., bulgaricus IFO13953	ARHPHPLSFM	Antioxidant
*Kluyveromyces marxianus*	Tyr-Leu-Leu-Phe YLLF	ACE inhibitor
β-Casein-derived peptides	Lys-Val-Leu-Pro-Val-P(Glu) KVLPVP(E)	ACE inhibitor

^a^ From Sánchez and Vázquez (2017) [12]. ^b^ ACE, Angiotensin Converting Enzyme (EC 3.4.15.1).

**Table 2 nutrients-14-04400-t002:** Previously described bioactive peptides with anti-allergic effects. ↑ Increase; ↓ decrease.

Source/Allergen	Peptide Sequence	Activity	Reference
Eggs/Ovalbumin (OVA)	- AMVYLGAKDSTRTQ - SWVESQTNGIIRNVL - AAHAEINEAGREVVG	↓Symptoms ↓Histamine ↓OVA-specific IgE ↑OVA-specific fecal IgA	[39]
Eggs/Ovomucoid (OVM)	- DNKTYGNKSNFSNAV	↓Symptoms ↓Histamine ↓OVM-specific IgE, ↑IgG1, ↓IgG2a, ↑fecal IgA ↓IL-4, ↑IL-12, ↑IL-10 released by OVM splenocytes	[40]
Milk/β-lactoglobulin (BLG)	- AQKKIIAEKTKIPAVFKIDALN - ALKALPMHIRLSFNP	↓Symptoms and temperature No change in BLG-specific IgE, IgG1, IgG2 or fecal IgA ↑IFN-γ, ↑IL-12, ↑IL-10 released by BLG splenocytes	[41]
Milk/Casein	- HAQ	↓inflammatory cytokines (IL-4) ↓Anaphylaxis-like symptoms	[42]
Pooled sera of allergen patients	- LSYLLWRSRLP - LVAHVGAGGVL - RVSSCRGRNHIV - ETIGARWVRIE - TDGVTYTNDCL - RVVRYDADFWI - GFWCRRSGLVGV	↓histamine, ↓calcium influx, ↓β- hexosaminidase, ↓phosphorylation of extracellular regulated kinase (ERK)	[43,44]
Synthesis	- RTY	↓mast cell degranulation and release of β-hexosaminidase	[43,45]
*Spirulina maxima*	- LDAVNR - MMLDF - ADSDGK	↓Histamine ↓intracelular Ca^2+^	[46,47,48]
Mollusk/Abalone intestine	- PFNQGTFAS	↓histamine, ↓PCA ↓inflammatory cytokines (TNF-α, IL-1β and IL-6)	[49]
Fish/Atlantic salmon byproduct	- TPEVHIAVDKF	↓β-hexosaminidase for IgE-mediated RBL-2H3 cell	[50,51]

**Table 3 nutrients-14-04400-t003:** Several bioactive peptides exhibit anti-inflammatory activity. ↑ Increase; ↓ decrease.

Source	Peptide Sequence	Model	Activity	Reference
Sturgeon muscle	- KIWHHTF - VHYAGTVDY - HLDDALRGQE	LPS-stimulated RAW264.7 cells	↓MAPK pathway ↓inflammatory mediators (NO, IL-6, and IL-1β) ↑SOD activity ↓MAPKS phosphorylation	[59]
*Salmon salar* skin	- APD - QA - KA - WG	Macrophages from RAW264.7 cells	↓NO, IL-6, IL-1β, and TNF-α	[60]
Salmon pectoral fit	-PAY	LPS-stimulated RAW264.7 cells	↓NO and PGE2 ↓inflammatory cytokines (TNF-α, IL6 and IL1β)	[61]
Juice of cooked tuna	PRRTRMMNGGR	LPS-stimulated RAW264.7 cells	↓inflammatory cytokines TNF-α, IFN-γ, and IL-2	[62]
*Meretrix meretrix* clams	NPAQDC	Macrophages from RAW264.7 cells	↓(COX)-2 activation ↓Pro-inflammatory cytokines ↓NO production	[63]
In-vitro-digested human milk and pooled intestinal samples from 8 infants fed human milk	13 peptides	LPS-treated human immune THP-1 macrophages	↓TNF-α and IL-8	[64]
Milk casein	QEPVL	Lymphocytes from male Balb/c mice	↓NO production ↓cytokines IL-4, IL-10, IFN-γ, and TNF-α	[65]
Gastrointestinal digestates of common bean milk and yogurt	γ-E-S-(Me)C γ-EL LLV	Basolateral EA.hy926 cells as shown cascades	↓TNF-α induced pro-inflammatory mediators of the nuclear factor κB (NF-κB) and mitogen-activated protein kinase (MAPK) signal	[66]
Simulated gastrointestinal digestion of extruded adzuki bean protein	KQSESHFVDAQPEQQQR	LPS-induced RAW 264.7 macrophages	↓production of IL-1, IL-6, TNF-α, and MCP-1.	[67]
Milk casein	VPP IPP	ApoE knockout mice	↓production of IL-6, IL-1β ↓expression of NF-κB-related genes, CD40, LCK, PIK3CG, IL1B, and MAP2K7	[68]
Milk casein	VPP IPP	VPP Murine preadipocyte cell line 3T3-F442A	↑Upregulated PPARg and adiponectin expression; ↓adipokine levels and NF-κB pathway	[69]

**Table 4 nutrients-14-04400-t004:** Principal in silico methods include databases, online tools, and software.

Category	Name	Website	Function
Protein database	NCBI Protein database	https://www.ncbi.nlm.nih.gov/	Basic sequence information for proteins
	UniProtKB	http://www.uniprot.org/	Basic sequence and structural information for proteins
	BIOPEP	http://www.uwm.edu.pl/biochemia/index.php/en/biopep	Protein sequence database
	RCSB Protein Data Bank	https://www.rcsb.org/pdb/home/home.do	
	PepBank	http://pepbank.mgh.harvard.edu/	
	MilkAMP	http://milkampdb.org/	
	PeptideDB	http://www.peptides.be/	
	AMPer	http://marray.cmdr.ubc.ca/cgi-bin/amp.pl	
	BioPD	http://biopd.bjmu.edu.cn/	
	SwePep	http://www.swepep.org/	
	EROP-Moscow	http://erop.inbi.ras.ru/	
In silico digestion tools	Peptide Cutter	http://web.expasy.org/peptide_cutter/	Server for predicting potential cleavage sites cleaved by proteases or chemicals in a given protein sequence
	BIOPEP	http://www.uwm.edu.pl/biochemia/index.php/en/biopep	Server for predicting potential cleavage sites cleaved by proteases in a given protein sequence
	Enzyme Predictor	http://bioware.ucd.ie/~enzpred/Enzpred.php	Tool to evaluate the evidence for which enzymes are most likely to have cleaved a sample containing peptides from hydrolyzed proteins
	Bioactive peptide database BIOPEP	http://www.uwm.edu.pl/biochemia/index.php/en/biopep	Bioactive peptide database
	BitterDB	http://bitterdb.agri.huji.ac.il/bitterdb/	Bitter compounds database
	EROP-Moscow database	http://erop.inbi.ras.ru	Database of biologically active peptides
	APD	http://aps.unmc.edu/AP/main.html	Several types of bioactive peptide databases with the main focus on antimicrobial peptides
	PeptideDB	http://www.peptides.be/	Biologically active peptide database
	PepBank	http://pepbank.mgh.harvard.edu/	Biologically active peptide database providing a search program for fragments with sequence similar to the peptides in thedatabase
	POPS	http://pops.csse.monash.edu.au/pops-cgi/index.php	
	AHTPDB	http://crdd.osdd.net/raghava/ahtpdb/	Antihypertensive peptide database
Potential bioactivity prediction	BIOPEP	http://www.uwm.edu.pl/biochemia/index.php/en/biopep	Tool for the evaluation of proteins as the precursors of bioactive peptides
	PeptideRanker	http://bioware.ucd.ie/~compass/biowareweb/Server_pages/peptideranker.php	Server for the prediction of bioactive peptides.
	PeptideLocator	http://bioware.ucd.ie/	
	AntiBP2	http://crdd.osdd.net/raghava//antibp2/	Predicting the antibacterial peptides in a protein sequence
Allergenicity/toxicity prediction/analyzing	AlgPred	http://crdd.osdd.net/raghava//algpred/	Predicting allergenic proteins and peptides
	BIOPEP	http://www.uwm.edu.pl/biochemia/index.php/en/biopep	Allergenic protein database
	ToxinPred	http://crdd.osdd.net/raghava//toxinpred/	Predicting toxicity of peptides
Physicochemical characteristics prediction	Expasy-Compute pI/Mw	http://web.expasy.org/compute_pi/	Tool to compute the theoretical pI (isoelectric point) and Mw (molecular weight)
	ProtParam	http://web.expasy.org/protparam/	Tool to compute grand average of hydropathicity (GRAVY) and instability index
	PepDraw	http://www.tulane.edu/~biochem/WW/PepDraw/	Tool to compute net charge and hydrophobicity
Peptide Structure Prediction	Server Pep-Fold	http://bioserv.rpbs.univ-paris-diderot.fr/services/PEP-FOLD/	Tool to predict peptide structures from amino acid sequences
	PEPstrMOD	http://osddlinux.osdd.net/raghava/pepstrmod	Server to predict the tertiary structure of small peptides
Protein Structure Prediction	Server I-TASSER	https://zhanglab.ccmb.med.umich.edu/I-TASSER/	Protein structure and function prediction
Mainly Software for the study of protein—ligand interactions	Discovery studio		
	Sybly		
	Autodock vina		
	Schrodinger		
	Dock		
	FlexX		
	ICM-Docking		
	GOLD		
	I-TASSER	https://zhanglab.ccmb.med.umich.edu/I-TASSER/	Protein structure and function prediction
In silico tools for molecular docking of peptides	DOCK Blaster	http://blaster.docking.org/	
	1-CLICK DOCKING	https://mcule.com/apps/1-click-docking/	
	BSP-SLIM	https://zhanglab.ccmb.med.umich.edu/BSP-SLIM/	
	SwissDock	http://www.swissdock.ch/	
	FlexPepDock	http://flexpepdock.furmanlab.cs.huji.ac.il/	
Identification and characterization of peptides, including tools for chemometrics	PubChem	https://pubchem.ncbi.nlm.nih.gov/	
	ProtParam	https://web.expasy.org/protparam/	
	FooDB	http://foodb.ca/	
	Chemical Entities of Biological Interest (ChEBI)	https://www.ebi.ac.uk/chebi/	
	AAindex	http://www.genome.jp/aaindex/	
	Human Metabolome Database (HMDB)	http://www.hmdb.ca/	
	Peptigram	http://bioware.ucd.ie/peptigram/	
	METLIN	https://metlin.scripps.edu/	

## Data Availability

Not applicable.

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
