# Peer review of "Proteomics Characterization of Food-Derived Bioactive Peptides with Anti-Allergic and Anti-Inflammatory Properties"

_nutrients, 2022, doi:10.3390/nu14204400_

Round 1

Reviewer 1 Report

A comprehensive and well-written review about bioactive peptides found in food. The review is focused on peptides derived mainly from proteolytic processing. It discusses different sources of peptides, their isolation and characterization techniques, and approaches to mining prospective peptides with biological activity. The manuscript is specially focused on anti-inflammatory and anti-allergenic peptides, though other potential pharmacological properties could be discovered. 

A few points should be revised:

- figure 1 needs some edits: check the atom bonds and sites S’1, S’2…

- on page 4 is written ‘These scientific methods, although satisfactory at a small scale, such as in laboratory procedures, may not be suitable for optimizing for commercial production, which occasionally requires the development of new procedures.’ 

This sentence is a little bit obscure. Could the authors please elaborate more and discuss these new procedures? 

- on page 7: ‘Using a combinatorial chemistry approach, we identified different bioactive peptides…’ and then is cited the reference 41. But I cannot find any of the authors of this manuscript in this reference. 

- check the references: some of them have ‘page 32’ like references 2, 3, and 13… and in others, like 76, it is not shown in the journal. 

Author Response

Thank you very much for the suggestions and the work. These have been taken into consideration in the revised form. 

Reviewer 2 Report

Solid work.

Comprehensible way of quoting the data.

Worth publishing

Just a few remarks.

 Line 45:        Perhaps for a scientific paper definition of nutraceuticals from other scientific sources would be preferable.

Line 103:       Please omit the word “whey”.

Line 115:       “It is currently clear that diseases ……….. inflammatory peptides”

Please add reference.

Line 134:              These authors, after an…….

                              Please name the authors.

Author Response

(The authors gave the same response as above.)

Reviewer 3 Report

Abril et al reviewed bioactive peptides with anti-allergic and anti-inflammatory properties and how they can be characterized using proteomics. In general, the article may serve as a source of information for scientists working in the relevant field. I have following minor comments:

1.     Figure 1: subscripts in figure 1 upper part needs to be fixed.

2.     Please provide a proper citation for this claim on page 4 line 115-117: “It is currently clear that diseases, such as type 2 diabetes, can be successfully treated simply by consuming diets that generate these bioactive anti-inflammatory peptides.”

3.     Page 12 line 420: I assume the author meant high sensitivity, please correct.

Author Response

(The authors gave the same response as above.)
